# Study on Spatial Geometric Similarity Based on Conformal Geometric Algebra

**DOI:** 10.3390/ijerph191710807

**Published:** 2022-08-30

**Authors:** Xiaomin Jiang, Yangfei Huang, Feng Zhang

**Affiliations:** 1School of Civil Engineering & Architecture, Zhejiang University of Science & Technology, 318 Liuhe Road, Hangzhou 310023, China; 2Center of Urban and Rural Development, Zhejiang University of Science & Technology, 318 Liuhe Road, Hangzhou 310023, China; 3Zhejiang Provincial Key Laboratory of Geographic Information Science, Zhejiang University, 148 Tianmushan Road, Hangzhou 310028, China; 4School of Earth Sciences, Zhejiang University, 38 Zheda Road, Hangzhou 310027, China

**Keywords:** conformal geometric algebra, shape similarity, spatial relations similarity, spatial geometric similarity

## Abstract

The study of spatial geometric similarity plays a significant role in spatial data retrieval. Many researchers have examined spatial geometric similarity, which is useful for spatial analysis and data retrieval. However, the majority of them focused on objects of the same type. Methods to support the spatial geometric similarity computation for different types of objects are rare, a systematic theory index has not been developed yet, and there has not been a comprehensive computational model of spatial geometric similarity. In this study, we conducted an analysis of the spatial geometric similarity computation based on conformal geometric algebra (CGA), which has certain advantages in the quantitative computation of the measurement information of spatial objects and the qualitative judgment of the topological relations of spatial objects. First, we developed a unified expression model for spatial geometric scenes, integrating shapes of objects and spatial relations between them. Then, we established a model for the spatial geometric similarity computation under various geographical circumstances to provide a novel approach for spatial geometric similarity research. Finally, the computation model was verified through a case study. The study of spatial geometric similarity sheds light on spatial data retrieval, which has scientific significance and practical value.

## 1. Introduction

With the diversification of Earth observation technology and the popularization of mobile devices, the amount and types of data related to geospatial positions are increasing rapidly. Determining how to locate spatial geometric scenes in which users are interested from massive spatial geometric scene data is a major issue for spatial data retrieval in urban scenarios. Computing the geometric similarity between a database scene and a query scene in a spatial database is a key point in solving the problem of spatial scene retrieval. The traditional spatial geometric similarity indices are the shape similarity of spatial objects, topological relation similarity, direction relation similarity, and distance relation similarity between spatial objects, and the computation methods are not uniform. The existing spatial data retrieval methods that were developed without comprehensive consideration of the above similarity indices cannot satisfy the demand for spatial geometric scene retrieval.

In 2001, Chinese mathematician Li Hongbo proposed the theory of conformal geometric algebra (CGA), and it was introduced to the geoscience field. This theory can realize the unified expression of different types and dimensions of geographic objects and has certain advantages in the quantitative computation of the measurement information of spatial objects and the qualitative judgment of the topological relations of spatial objects [1].

In this study, we focused on the problem of spatial geometric similarity. Based on CGA theory, a discussion is presented from the perspective of the shape expressions of spatial objects and similarity computation, the expression of the spatial relations and similarity computation, and the computation of the geometric similarity of a spatial scene. Finally, a set of relatively complete spatial geometric similarity retrieval models is proposed, which can satisfy the demand of qualitative spatial data retrieval to a certain degree.

The main contributions of the present study are as follows.

(1)Based on CGA, we realize a unified representation of spatial geometric scenes with objects of different types. The spatial geometric scene we consider is composed of different types of geometric objects and is not restricted to polygons.(2)We perform geometric similarity computations between two spatial geometric scenes, taking the shape, topology, direction, and distance into consideration to attempt to implement spatial geometric retrieval in a comprehensive manner.

## 2. Literature Review

A spatial scene comprises a collection of spatial objects and their particular spatial arrangement. People construe such spatial geometric scenes in terms of objects and relations [2,3]. Many scholars have explored this topic [4,5,6,7]. Spatial scene similarity is the similarity of two regions in terms of particular object sizes and contents [5,6,7,8]. The spatial geometric similarity in our study involves spatial geometric features, including the shape of spatial objects and the spatial relationship between objects. The existing spatial similarity research is mainly divided into two parts. One is based on the geometric characteristics of spatial objects, and the similarity of spatial objects is studied. Kuijpers et al. [9] and Gottfried [10] used a double cross model and bipartite arrangements, respectively, to obtain qualitative shape expressions and proposed concrete qualitative shape similarity computation algorithms from the perspective of shape description operations. Shapiro et al. [11] proposed a theoretical relational modeling technique to separate objects into thin pieces (e.g., sticks, plates, and blobs). Richards et al. [12] used a codon as the basic unit to describe closed two-dimensional (2D) shapes.

The other approach uses the relations between spatial objects to study the similarity of the spatial relations [13,14,15]. Spatial relation similarity mainly includes spatial topological relation similarity, direction relation similarity, and distance relation similarity. Many scholars have performed some research on the combination of topology and direction, the combination of direction and distance, and the combination of the three and developed similarity evaluation methods, such as feature matching, derivation from equivalence, case reasoning, and fuzzy functions [16,17,18,19,20,21]. The semantic similarity plays an important role in spatial scene retrieval. Many researchers have made some contributions in this area [22,23,24,25].

Geometric algebra is an algebraic language that was proposed by Clifford [26] based on the Hamilton Quaternion and Grassmann expansion algebra to describe and perform computations for geometric problems. The unification of vector, scalar, dimension, and geometric operations makes it possible to apply geometric algebra for the expression of geographic objects and compute the spatial relations. The research on the application of geometric algebra has matured around the world [27,28,29,30], and there are various research results in the theoretical physics [31,32], mathematics [33,34], computer science [35], and geoscience [36,37,38,39,40,41,42,43,44,45,46,47,48] fields.

Li et al. [1] first presented the concept of CGA in the book Geometric Computing with Clifford Algebras, which has been highly praised by the international academic community. Zong [47] obtained a formal expression of the spatial topological relations based on algebraic geometry and the quantitative computation of topological relations by using a meet operator, which was able to distinguish more detailed spatial topological relations, and applied it to the intersection of the 3D surface model of an Antarctic ice sheet. Chen [48] obtained a unified expression of different types of multi-dimensional geographic objects using CGA properties, based on which formal expressions of the spatial topological relations and temporal topological relations were obtained. Furthermore, spatiotemporal topological relations were obtained. These relations were applied to the computation of spatiotemporal topological relations. This approach has value for various applications, such as checking the spatiotemporal topological rules of current land use and the historical backtracking of spatiotemporal objects.

## 3. Methodology

In this paper, we introduce a multidimensional unified expression and geometric computation based on CGA. The unified expression of multidimensional geometric objects is realized based on the multivector. By using the geometric product, the similarity relationship of spatial objects, spatial topology, direction, and distance can be conveniently computed, which provides a new solution for the spatial geometric similarity retrieval.

According to Figure 1, first, we resolve the central objects that satisfy a specified shape similarity threshold. Second, the Voronoi graph is built to obtain the spatial neighborhood objects. From this, we can obtain a certain number of scenes. Third, according to the spatial constraints for the query scene, the variable ordering is set, and the forward checking algorithm is used to match the query scene with the sub-scenes from the database scene. If no such sub-scene meets the condition, we can relax the condition and compute the similarity between the query scene and the retrieved scene until we achieve our goal. The main theories covered in this paper are described in this section as follows.

### 3.1. Conformal Geometric Algebra

CGA is a new geometric representation and computing system that provides a unified and simple homogeneous algebraic framework for classical geometry and has notable advantages in geometric data processing and geometric computation [49]. The inner product, outer product, and geometric product are core operations in the geometric object representation and algorithm construction. CGA is the most common geometric algebra (GA) space. Based on the traditional Euclidean space, two extra base vectors, *e*_0_ and *e_i_*, are used to represent basic geometric objects. A hierarchical geometric structure corresponds to the hierarchical Grassmann structure for the outer product in CGA. The inner product in CGA can describe basic metric information, such as the distance and angle.

### 3.2. Shape Similarity Computation

Falomir et al. [50] conducted similarity measurements of spatial objects based on qualitative shape descriptions. The complete description of the shape of a 2D object is given from a set of qualitative features as follows:$$\left[EC_{1},A_{1}|TC_{1},L_{1},C_{1}],[EC_{2},A_{2}|TC_{2},L_{2},C_{2}],\ldots ,[EC_{n},A_{n}|TC_{n},L_{n},C_{n}\right]$$
where *n* is the total number of relevant points of the object, *EC_i_* describes the edge connection that occurs at the point *P_i_*, *A_i_*|*TC_i_* describes the angle or the type of curvature at *P_i_*, *L_i_* describes the compared length of the edges connected at *P_i_*, and finally, *C_i_* describes the convexity at *P_i_*.

In this study, the qualitative shape representation model (QSDM) of spatial objects is established on the basis of their research. The spatial objects are edge-connected, so we employ the other three features.

As shown in Figure 2, it is assumed that *P*_2_, *P*_3_, and *P*_4_ are three points of the spatial object *S*, which are connected in turn. At point *P_3_*, a three-tuple is defined to represent the relevant point,
(1)$$\langle A_{P},L_{P},C_{P}\rangle ,$$
where *A_P_* denotes the angle between line segments *P*_2_*P*_3_ and *P*_3_*P*_4_, *L_P_* denotes the ratio of the length of the line segments, and *C_P_* denotes the concavity and convexity of point *P*_3_.

#### 3.2.1. Angle Computation

The angle between two adjacent line segments of the polygon is computed at relevant points by using the inner product. For one-dimensional vectors *a* and *b*, which are constructed by connecting a relevant point *P*_3_ to its previous point *P*_2_ and its next point *P*_4_, respectively, the angle is computed by GA, which encodes the inner and the outer product; the inner product is proportional to the cosine, and the outer product is the sine. Thus, the value of the angle can be computed by the arctan2 function. At first, we partition the angle into five intervals, {[0, 40], (40, 85], (85, 95], (95, 140], (140, 180]}, and the corresponding semantic expressions of the five intervals are {very_acute, acute, right, obtuse, very_obtuse}. In this work, we use an open ball *B_r_(c)* (Borelian notation) where $c=\left(a+b\right)/2$(center) and $r=\left(b-a\right)/2$(radius). Given two internals, $I_{1}=\left(a_{1},b_{1}\right)=B_{r1}\left(c_{1}\right)$ and $\;I_{2}=\left(a_{2},b_{2}\right)=B_{r2}\left(c_{2}\right)$, a family of distances between intervals was defined [51], which depends on three parameters as follows:(2)$$d^{2}\left(I_{1},I_{2}\right)=\left(\Delta c\;\Delta r\right)A\left(\begin{array}{c} \Delta c\\ \Delta r \end{array}\right),$$
where $\Delta c=c_{2}-c_{1}$, $\Delta r=r_{2}-r_{1}$, and *A* is a symmetrical 2 × 2 matrix of weights. Here, we use the most natural choice for the *A* matrix, which is the identity matrix that provides the next distance:(3)$$d^{2}\left(I_{1},I_{2}\right)=\sqrt{\Delta^{2}c+\Delta^{2}r}=\sqrt{{\left(c_{2-}c_{1}\right)}^{2}+{\left(r_{2-}r_{1}\right)}^{2}}.$$

Using Borelian notation and the interval distance, for example, for the two intervals [0, 40] and [40, 85], $I_{1}=\left(0,40\right)=B_{20}\left(20\right)$ and $I_{2}=\left(40,85\right)=B_{22.5}\left(62.5\right)$, $d^{2}\left(I_{1},I_{2}\right)=\sqrt{{\left(62.5-20\right)}^{2}+{\left(22.5-20\right)}^{2}}$ = 42.6, which is the distance between the very acute and acute. Using the same process, we obtain all the angle relation distances, as shown in Table 1.

#### 3.2.2. Ratio of Length Computation

The lengths of two adjacent line segments of polygon feature points are computed using the inner product. The Euclidean distance between two points is directly embedded in the inner product computation with the expression $p_{1}\cdot p_{2}=-\frac{1}{2}d_{E}^{2}\left(p_{1},p_{2}\right)$, and the ratio of the lengths is the ratio of the lengths of two adjacent line segments.

We partition the ratio of the lengths into seven intervals, {(0, 0.4], (0.4, 0.6], (0.6, 0.9], (0.9, 1.1], (1.1, 1.9], (1.9, 2.1], (2.1, 10]}, and the corresponding semantic expressions of the seven intervals are {much_shorter (msh), half_length (hl), a_bit_shorter (absh), similar_length (sl), a_bit_longer (abl), double_length (dl), much_longer (ml)}. Using Borelian notation and the interval distance, the computing method is the same as Formula (3), and we obtain the angle relation distances, as shown in Table 2.

#### 3.2.3. Concavity and Convexity Judgment

The concavity and convexity of the angle between two adjacent line segments of the polygon are judged using the inner product. For example, for point *P*_2_, its concavity and convexity can be determined by the relation between a one-dimensional vector *a* = *P*_1_*P*_2_ and the point *P*_3_. If the point *P_3_* is on the left of vector *a*, the point *P*_2_ is concave; otherwise, *P*_2_ is convex. The concrete computation can construct a one-dimensional vector *c* = *P*_1_*P*_3_ and a one-dimensional vector *m* that is perpendicular to *P_1_P_2_* on the right, and then the sign of the inner product *c·m* is examined. If *c·m* > 0, point *P_3_* is on the right of vector *a*. If *c·m* < 0, point *P*_3_ is on the left of vector *a*. For concavity relation distance, it equals 0 if they are the same; otherwise, it equals 1. For two concave or convex relations, the distance is 0, and for one concave and the other convex, the distance is 1.

The computations of the three parameters above are the components of the shape computation, which mainly rely on inner product computation. An inner product contains the information of the angle and distance, which can obtain the three-tuple quantitative values of the shape expression model concisely and intuitively and lay the foundation for the computation of the shape similarity.

For two objects, we adopt the following formula to compute the shape similarity between them:(4)$$Sim(RP_{A},RP_{B})=1-\sum_{i\in \{A,L,C\}}w_{i}\frac{ds(i)}{Ds(i)},$$
where *RP_A_* and *RP_B_* denote the relevant points of objects *A* and *B*, respectively; *ds*(*i*) denotes the distance between two relations; *Ds*(*i*) denotes the maximum distance; and *w_i_* denotes weights for each relation.

### 3.3. Topological Relations Similarity Computation

Zong [47] performed research on spatial topological relations based on geometric algebra and discussed the topological relations between a point, line segment, and triangle. Chen [48] sought the topological relations between two polygons by decomposing a polygon into triangles. In this article, we do not distinguish the topological relations as Chen did, but rather, we take the polygon as a whole.

The spatial topological relations are determined based on the research of Zong and Chen. We divide the object into the interior, boundary, and exterior (the point can be divided into the interior and the exterior, without the boundary), use the geometric product, and construct different discriminants. Thus, topological relations between objects of different types can be obtained.

#### 3.3.1. Point and Other Objects

(1)Topological relations between two points

For two points *A* and *B*, the discriminant is constructed as follows:(5)t=A∧B.

If *t* = 0, *A* intersects *B*, and the topological relation is pp-equal. If *t* <> 0, *A* does not intersect *B*, and the topological relation is pp-disjoint. The topological relations are shown in Figure 3.

Using the topological relation operation between two points, we obtain two relations: pp-disjoint and pp-equal. Figure 4 shows the corresponding conceptual neighborhood diagram. For two points, when the topological relation is the same as the other two, the relation distance is 0; otherwise, the relation distance is 1.

(2)Topological relations between point and line segment

A line segment is a straight line with a boundary constraint. For a straight line *L* passing through points *A* and *B*, the judgment expression is *L* = *A ^ B ^ e_i_*. The topological relations between point *P* and line segment *AB* can be determined by the two following expressions:(6)$$t_{1}=(A\wedge P\wedge e_{i})L,\;t_{2}=(P\wedge B\wedge e_{i})L.$$

If *t*_1_ > 0 and *t*_2_ > 0, *P* is within *AB*, and the topological relation between them is pl-inside. If *t*_1_ = 0 or *t*_2_ = 0, *P* coincides with the edges of *AB*, and the topological relation between them is pl-touch. If *t*_1_ < 0 or *t*_2_ < 0, *P* does not intersect with *AB*, and the topological relation between them is pl-disjoint. The topological relations are shown in Figure 5.

Using the topological relation operation between a point and a line segment, we obtain three relations: pl-disjoint, pl-touch, and pl-inside. Figure 6 shows the corresponding conceptual neighborhood diagram. The topological relation distances which are decided according to the conceptual neighborhood diagram are shown in Table 3.

(3)Topological relations between point and polygon

The relations can be computed using the following judgment expression:(7)$$t=\left(P\wedge L\right)\cdot \Phi ,$$
where *P* is the point, *L* is the boundary of the polygon, and Φ is the plane that contains the polygon *M*. $\Phi =C\wedge e_{i}$, *C* is the boundary circle, *e_i_* denotes infinity, and *L* denotes the straight line at which the boundary of the polygon *M* is located.

If *t* < 0 for all judgment expressions, *P* is inside *M*. The topological relation between them is ps-inside. If *t* = 0, *P* is in the boundary of *M*. The topological relation between them is ps-touch. In other situations, *P* is outside *M*. The topological relation between them is ps-disjoint. The topological relations are shown in Figure 7.

Using the topological relation operation between a point and a polygon, we obtain three relations: ps-disjoint, ps-touch, and ps-inside. Figure 8 shows the corresponding conceptual neighborhood diagram. The topological relation distances which are decided according to the conceptual neighborhood diagram are shown in Table 4.

#### 3.3.2. Two Line Segments

Using the operation between a point and a line segment, we can construct a computing strategy to determine the topological relations between two line segments. For two line segments *L*_1_ and *L*_2_, the relation between the two edge points of *L*_1_ and the line segment *L*_2_ is computed. When one of the edge points of *L*_1_ is within *L*_2_ or coincident with the edge points of *L*_2_, the topological relation is ll-touch. When the two edges of *L*_1_ coincide with two edges of *L*_2_, the topological relation is ll-equal. When the two edge points of *L*_1_ are inside *L*_2_ and the two edge points of *L*_2_ are outside *L*_1_, the topological relation is ll-inside. When the two edge points of *L*_2_ are inside *L*_1_ and the two edge points of *L*_1_ are outside *L*_2_, the topological relation is ll-contains. When the interior of *L*_2_ intersects with that of *L*_1_, the topological relation is ll-overlap. When both the interior and the two edge points of *L*_2_ do not intersect with the interior and the two edge points of *L*_1_, the topological relation is ll-disjoint. The topological relations are shown in Figure 9.

Using the topological relation operation between two line segments, we obtain six relations: ll-disjoint, ll-touch, ll-overlap, ll-equal, ll-contains, and ll-inside. Figure 10 shows the corresponding conceptual neighborhood diagram. The topological relation distances which are decided according to the conceptual neighborhood diagram are shown in Table 5.

#### 3.3.3. Line Segment and Polygon

Using the operation between a point and a polygon and the operation between two line segments, we compute the topological relations as follows. For the two edge points of *L*, if both are inside polygon *S*, the topological relation between *L* and *S* is ls-inside1. If one edge point of *L* is inside polygon *S* and the other edge point is on the boundary of *S*, the topological relation between *L* and *S* is ls-inside2. If the two edge points of *L* are on the boundary of *S*, the topological relation between *L* and *S* is ls-inside3. If one edge point of *L* is on the boundary of *S* and the other is outside *S*, the topological relation between *L* and *S* is ls-touch. If one edge point of *L* is inside *S* and the other is outside *S*, the topological relation between *L* and *S* is ls-overlap1.

If the two edge points of *L* are outside *S*, the relation between the line segment *L* and the boundary of *S* is judged. If *L* does not intersect with the boundary of *S*, the topological relation between *L* and *S* is ls-disjoint. If *L* intersects with the boundary of *S*, the topological relation between *L* and *S* is ls-overlap2.

In summary, we can distinguish four topological relations: ls-disjoint, ls-overlap, ls-inside, and ls-touch. The relations ls-inside1, ls-inside2, and ls-inside3 can be recognized as ls-inside, and the relations ls-overlap1 and ls-overlap2 can be recognized as ls-overlap. The topological relations are shown in Figure 11.

Using the topological relation operation between a line segment and a polygon, we obtain four relations: ls-disjoint, ls-touch, ls-overlap, and ls-inside. Figure 12 shows the corresponding conceptual neighborhood diagram. The topological relation distances which are decided according to the conceptual neighborhood diagram are shown in Table 6.

#### 3.3.4. Two Polygons

Based on the operation between a point and a line segment, between a point and a polygon, and between two line segments, we compute the topological relations as follows.

For all vertices {*U*_1_, *U*_2_, …, *U_m_*} of *S*_1_, if all are inside *S*_2_, the operation between a point and a polygon is used, and the condition is recorded as *C*_1_. If all the vertices are outside *S*_2_, the condition is recorded as *C*_2_. If they are all on the boundary of *S*_2_, the condition is recorded as *C*_3_. If some of the vertices are inside *S*_2_ and some are outside *S*_2_, the condition is recorded as *C*_4_. If some of the vertices are inside *S*_2_ and some are on the boundary of *S*_2_, the condition is recorded as *C*_5_. If some of the vertices are outside *S*_2_ and some are on the boundary of *S*_2_, the condition is recorded as *C*_6_. Likewise, we obtain the conditions *D*_1_, *D*_2_, *D*_3_, *D*_4_, *D*_5_, and *D*_6_.

If *C*_1_ is satisfied but *D*_1_ is not, the topological relation between *S*_1_ and *S*_2_ is ss-contains. If *D*_1_ is satisfied but *C*_1_ is not, the topological relation between *S*_1_ and *S*_2_ is ss-inside. If both *C*_2_ and *D*_2_ are satisfied, the topological relation between *S*_1_ and *S*_2_ is ss-disjoint. If both *C*_3_ and *D*_3_ are satisfied, the topological relation between *S*_1_ and *S*_2_ is ss-equal. If *C*_4_ or *D*_4_ is satisfied or both are, the topological relation between *S*_1_ and *S*_2_ is ss-overlap. If *D*_5_ is satisfied, the topological relation between *S*_1_ and *S*_2_ is ss-covers. If *C*_5_ is satisfied, the topological relation between *S*_1_ and *S*_2_ is ss-covered-by. If both *C*_6_ and *D*_6_ are satisfied, the topological relation between *S*_1_ and *S*_2_ is ss-touch. The topological relations are shown in Figure 13.

Using the topological relation operation between two polygons, we obtain eight relations: ss-disjoint, ss-touch, ss-overlap, ss-covers, ss-covered-by, ss-contains, ss-inside, and ss-equal. Figure 14 shows the corresponding conceptual neighborhood diagram. The topological relation distances which are decided according to the conceptual neighborhood diagram are shown in Table 7. The topological relations between two polygons can be distinguished using the same topological relations with the RCC8 model.

#### 3.3.5. Topological Relation Similarity Computation Strategy

*m* spatial objects in query scene *V* are denoted as *v*_1_, *v*_2_, …, *v_m_*. *n* spatial objects in the query database *U* are denoted as *u*_1_, *u*_2_, …, *u_n_*. We adopt the following strategy to compute the topological relations between two scenes. First, the topological relation operation is used to resolve the relation between two objects in the query scene, which is denoted as *T*_1_. Furthermore, we obtain the relation between two objects in the database scene, which is denoted as *T*_2_. By utilizing the corresponding conceptual neighborhood diagram, the relation distance is computed. Furthermore, the similarity *S_Ri_* between *T*_1_ and *T*_2_ is computed.

We compute all topological relation similarities between objects in the query scene and corresponding objects in the database scene as follows:(8)$$S_{Rel}=\frac{\sum_{i=1}^{t\cdot \left(t-1\right)/2}W_{Ri}\cdot S_{Ri}}{\sum_{i=1}^{t\cdot \left(t-1\right)/2}W_{Ri}},$$
where *W_Ri_* denotes the weight values of the topological relations. In this paper, the weight values are the same; however, for future work, the weight values can be different according to the actual scene. For example, the topological relation inside may be more important than disjoint. *S_Ri_* denotes the similarity between two relations, and *t* denotes the number of matched objects.

### 3.4. Direction Relation Similarity Computation

The outer product realizes the dimensional and geometric operations, while the inner product provides the angle and distance information. For two geometric objects *O_1_* and *O_2_* of arbitrary dimensions, the metric information between two basic objects is as follows:(9)$$\cos(\theta )=\frac{o_{1}•o_{2}}{\left|o_{1}||o_{2}\right|},$$
(10)$$\theta =∠(o_{1},o_{2})=\arccos(\frac{o_{1}•o_{2}}{\left|o_{1}||o_{2}\right|}).$$

For example, as shown in Figure 15a, for two vectors *O*_1_ and *O*_2_, the inner product is the distance that *O*_1_ projects onto *O*_2_. In Figure 15b, for the vector *O*_1_ and the plane *O*_2_, the inner product is the vector that is in *O*_2_ and perpendicular to *O*_1_. Owing to high-dimensional scalability, the operation is conferrable to arbitrary dimensions, not limited to vectors [52].

If *P*_1_(0,1) and *P*_2_(1,0) belong to a 2D Euclidean space, by converting them into the conformal space, we obtain *C2ga_P*_1_ = *e*_2_ + 0.5*e**_∞_* + *e_0_* and *C2ga_P*_2_ = *e*_1_ + 0.5*e**_∞_* + *e_0_*. The origin point is *C2ga_O* = *e*_0_. The outer product between two points and infinity can be used to construct a line. Thus, the line *P_1_P_2_* can be represented as *C2ga_P*_1_ *^ C2ga_P*_2_ *^ e**_∞_*. The magnitude of *P*_1_*P*_2_ is *|P*_1_*P*_2_*|* =$\;\sqrt{2}$, and the magnitude of the unit base vector is *|OE|* = 1. Thus, the inner product *P*_1_*P*_2_*·OE* = *|P*_1_*P*_2_*||OE|*cosθ, and the computation result is $\;\cos\theta =-\sqrt{2}/2$, that is, θ=135°.

The conformal space is angle-preserving. The angle computed by the inner product in conformal space reflects the angle in Euclidean space. The cosine function was symmetric within the scope of [0°, 360*°*]. If y_OA_ − y_OB_ < 0, then $\theta \in \left[0,\;180{}^{\circ}\right]$. If y_OA_ − y_OB_ > 0, then $\theta \in \left[180{}^{\circ},\;360{}^{\circ}\right]$.

In CGA, according to the Grassmann hierarchical structure, the minimum bounding circle can be constructed by the vertices of the polygon. The radius of the circle is obtained by $r=\sqrt{\frac{-C^{2}}{{\left(C\wedge e_{i}\right)}^{2}}}$, and the center of the circle is computed by *O* = *Ce_i_C*.

Thus, for two geometric objects *A* and *B*, the direction relation operation is implemented as follows. (1) The center points *O_A_* and *O_B_* of the minimum bounding circles are computed for *A* and *B*, respectively. (2) The inner product between *O_A_O_B_* and the unit base vector are computed, and then the angle value can be computed.

The angle value can be mapped to the corresponding angle interval, and the corresponding semantic expression can be acquired. Figure 16 illustrates the relation between the direction relation semantic expression and the angle value.

The direction relation semantic expression is as follows: {N, NE, E, SE, S, SW, W, NW}, representing north, northeast, east, southeast, south, southwest, west, and northwest, respectively. The corresponding value intervals are {[67.5, 112.5), [22.5, 67.5), [337.5, 22.5), [292.5, 337.5), [247.5, 292.5), [202.5, 247.5), [157.5, 202.5), [112.5, 157.5)}. In Borelian notation, we obtain {B_22.5_(90), B_22.5_(45), B_22.5_(0), B_22.5_(315), B_22.5_(270), B_22.5_(225), B_22.5_(180), and B_22.5_(135)}. Using Formula (3), we find all r = 22.5, and the distance d = 45 between two adjacent directions. Hence, we simplify the distance, which is similar to the conceptual neighborhood diagram.

The direction relation is reflexive. When the direction relation distance surpasses 4, the format *Dir_Max_* − *Dir_Cal_* is used to compute the relation distance. The final direction relation distances are shown in Table 8.

We employ the direction relation operation to compute the angle value between two objects, map the angle value to an angle interval, obtain the semantic expression, and compute the distance and similarity of the direction relations. The similarity computation formula of the direction relation is as follows:(11)$$Sim(Dr_{0},Dr_{1})=1-\frac{dr}{Dr},$$
where *dr* denotes the direction relation distance and *Dr* denotes the maximal distance.

### 3.5. Distance Relation Similarity Computation

As discussed in Section 3.4, the inner product contains the angle and distance information. In this section, we compute the distance information. The inner product is *C2ga_P*_1_*·C2ga_P*_2_ = (*e*_2_ + 0.5*e**_∞_* + *e*_0_)(*e*_1_ + 0.5*e**_∞_* + *e*_0_). Based on the equations $e_{0}^{2}=e_{\infty }^{2}=0$ and $e_{0}\cdot e_{\infty }=-1$, the value of the inner product is −1. The Euclidean distance is embedded in the inner product based on the expression $a\cdot b=-\frac{1}{2}d_{E}^{2}\left(a,b\right)$, and thus, the Euclidean distance is $\sqrt{\left(-2\right)*\left(-1\right)}$ = $\sqrt{2}$.

For two geometric objects *A* and *B*, the distance relation operation is implemented as follows. (1) The center points *O_A_* and *O_B_* of the minimum bounding circles are computed for *A* and *B*, respectively. (2) The inner product between *O_A_* and *O_B_* is computed, and then the distance value can be computed.

In CGA, the distance relation operation is used to compute the distance between two objects. We determine the relation between the quantitative distance value and the distance interval and obtain the semantic distance expression. Using the distance relation table, we can compute the distance relation similarity.

In an actual scene, the distance value range is uncertain. We use the ratio of the specific value and maximal value to represent the distance relation. In this study, we define the distance relation semantics expression as {very close (vc), close (cl), a bit close (abc), commensurate (co), a bit far (abf), far (f), very far (vf)}. The corresponding distance intervals are {[0, 0.1), [0.1, 0.16), [0.16, 0.2), [0.2, 0.3), [0.3, 0.5), [0.5, 0.7), [0.7, ∞)}. In Borelian notation, we obtain {B_0.05_(0.05), B_0.03_(0.13), B_0.02_(0.18), B_0.05_(0.25), B_0.1_(0.4), B_0.1_(0.6), B_0.15_(0.85)}. The distance relation distances computed by using Formula (3) are shown as Table 9.

The similarity computation formula for the distance relation is as follows:(12)$$Sim(Dis_{0},Dis_{1})=1-\frac{ds}{Ds},$$
where *ds* denotes the relation distance and *Ds* denotes the maximal distance.

### 3.6. Spatial Geometric Similarity Computation

Based on the whole process shown in Figure 1, we designed a retrieval method taking the shape and spatial relations (including topological, direction, and distance relation) into account. During the whole process, we assume that the weights are equivalent for the objects and the spatial relations between them. The topology, direction, and distance relations have the same effect on the spatial relation.

The scenes in Figure 17 are considered as an example. We apply the object shape and spatial relation similarity computation model to analyze the example. For the query scene (a), using the multivector and corresponding relation operations, we obtain the following integrating representation [46] form:(13)$$\begin{array}{l} GS_{a}=\{\langle A,B\rangle +\langle A,C\rangle +\langle A,D\rangle +\langle A,E\rangle \\ +\langle B,C\rangle +\langle B,D\rangle +\langle B,E\rangle +\langle C,D\rangle +\langle C,E\rangle +\langle D,E\rangle \} \end{array}$$

The shape and spatial relation information are implied in the representation. The related operations including topological relation, direction relation, and distance relation can be called to compute the corresponding relations directly. The topological, direction, and distance relations in the query scene are listed in Table 10, Table 11 and Table 12, respectively.

For the database scene (b), following the same computation strategy, we obtain the topological, direction, and distance relations, as shown in Table 13, Table 14 and Table 15, respectively.

Based on the spatial relations computed by the corresponding operations, the topological, direction, and distance relation similarities between (a) and (b) are *S_Top_* = (0.75 × 1 + 1 × 9)/10 = 0.975, *S_Dir_* = (1 × 10)/10 = 1.0, and S*_Dis_*= ((1 − 0.255/0.806) × 2 + (1 − 0.158/0.806) + 7 × 1.0)/10 = 0.917, respectively. For (a) and (b), the attribute and shape are the same, so the spatial similarity is *Sim* (*a*, *b*) = 1/2 + (*S_Top_* + *S_Dir_* + *S_Dis_*)/6 = 0.968.

In the database scene (c), the different colors represent different attributes, and only object B was different. When taking the shape and spatial relations into consideration, the similarities between (c) and (a) are equal, and both are 1. For the database scene (d), we obtain the spatial relation tables shown in Table 16, Table 17 and Table 18.

Based on the spatial relations computed by corresponding operations, the topological, direction, and distance relation similarities between (a) and (d) are *S_Top_* = (1 × 10)/10 = 1.0, *S_Dir_* = ((1 − 0.25) × 3 + (1 − 0.75) × 3 + 1 × 4)/10 = 0.7, and *S_Dis_* = ((1 − 0.2/0.806) × 2 + (1 − 0.255/0.806) × 1 + (1 − 0.158/0.806) × 1 + 1.0 × 6)/10 = 0.899, respectively. The attributes and shapes of the two scenes are the same, so the shape similarity is 1. Hence, the spatial similarity is *Sim* (*a*, *d*) = 1/2 + (*S_Top_* + *S_Dir_* + *S_Dis_*)/6 = 0.933. According to the computation results, the spatial similarity between (b) and (a) is bigger than that between (d) and (a), so the scene (b) is more similar than (d) to (a).

For the database scene (e), the spatial relations are the same as in the query scene (a). The shape of the object *C* is different: it is a triangle in (a) and a square in (e). The shape similarity operation is used first to construct a three-tuple representation for the relevant points. For scene (a), the relevant point representations of object *C*_1_ are <acute, sl, convex>, <acute, sl, convex>, and <acute, sl, convex>. Similarly, the point representations of object *C*_5_ are <right, sl, convex>, <right, sl, convex>, <right, sl, convex>, and <right, sl, convex>. Thus, the shape similarity between *C*_1_ and *C*_5_ is *Sim* = ((1 − 32.6/(140.0 × 3)) × 3)/4 = 0.69. Thus, the spatial similarity between scenes (a) and (e) is *Sae* = 0.5 + ((0.69 + 4)/5)/2 = 0.969. The similarity values between each scene and the query scene are shown in Table 19.

## 4. Case Study

### 4.1. Related Background

For resident areas in a city, public transportation stations are common facilities along the roads. Public transportation stations are simplified as point objects, roads as polyline objects, and resident buildings as polygons. The resident areas are spatial geometric scenes with objects of different types. The traditional methods cannot satisfy the retrieval demands for these scenes very well. In this section, we attempt to obtain a unified representation and computation for spatial geometric scenes to overcome the disunity of the representation and complexity of the spatial relation computation.

### 4.2. Retrieval Process

(1)An integrating representation model of geographical objects and geographical scenes is developed based on CGA. The model contains the geometric and spatial relation information for the scene. We can call related operations to resolve the information. For example, one query scene is shown in Figure 18. The database scene is shown in Figure 19.

(2)Objects that satisfy the specified similarity threshold with the scene’s central objects are resolved. Three parameters, i.e., closeness, perimeter, and area, are employed to compute the central objects. The closeness computation formula is as follows:
(14)$$FormFactor=\frac{4\pi \times Area_{Object}}{Perimeter_{Object}^{2}}.$$
The normalization formula is as follows:(15)$$Val=\frac{Com_{A}-Com_{B}}{Max-Min}.$$
Reasonable weights for each normalized parameter are assigned, and their weighted sum is calculated as follows:(16)Valfinal=∑i=1nwValii.


(3)The Voronoi graph and corresponding sub-scenes based on the neighborhood relations are constructed.(4)The spatial geometric similarity computation method is used, and then the similarity between each sub-scene and the query scene is obtained. The retrieval results are shown in Figure 20.

The method proposed in this study can be used to retrieve spatial scenes with objects of different types. The retrieval results will be helpful for real estate developers and civilians to retrieve similar spatial geometric scenes. It is of great scientific significance and practical value to conduct spatial scene retrieval.

## 5. Conclusions

The amount of spatial geometric data in urban scenarios is increasing. In this study, we propose a method of computing the spatial geometric similarity in urban spatial patterns based on CGA. Operations for the shape and spatial relations are constructed using the inner, outer, and geometric products in CGA, and the relation distance tables are developed by means of conceptual neighborhood diagrams and interval distances. Then, the process of spatial geometric similarity computation is designed and realized based on the expression and computation of the object shape and spatial relations between two objects. From the aspects of the spatial object shape, spatial topology, direction, and distance between two objects, the spatial geometric similarity is evaluated more completely, which is important in the field of spatial data retrieval. In further research, on the one hand, semantic similarity should be taken into consideration for spatial similarity computation. The relations between the topology, direction, and distance can be deduced before the spatial similarity computation to improve the retrieval efficiency. On the other hand, we will adopt some computation strategies, such as parallel computing using MapReduce, to improve the efficiency. The method proposed in this paper can carry out comprehensive quantitative calculation by integrating spatial topology, direction, distance, and spatial object, which has important reference significance for urban planning and city site selection, providing a new perspective for spatial planning.

## Figures and Tables

**Figure 1 ijerph-19-10807-f001:**
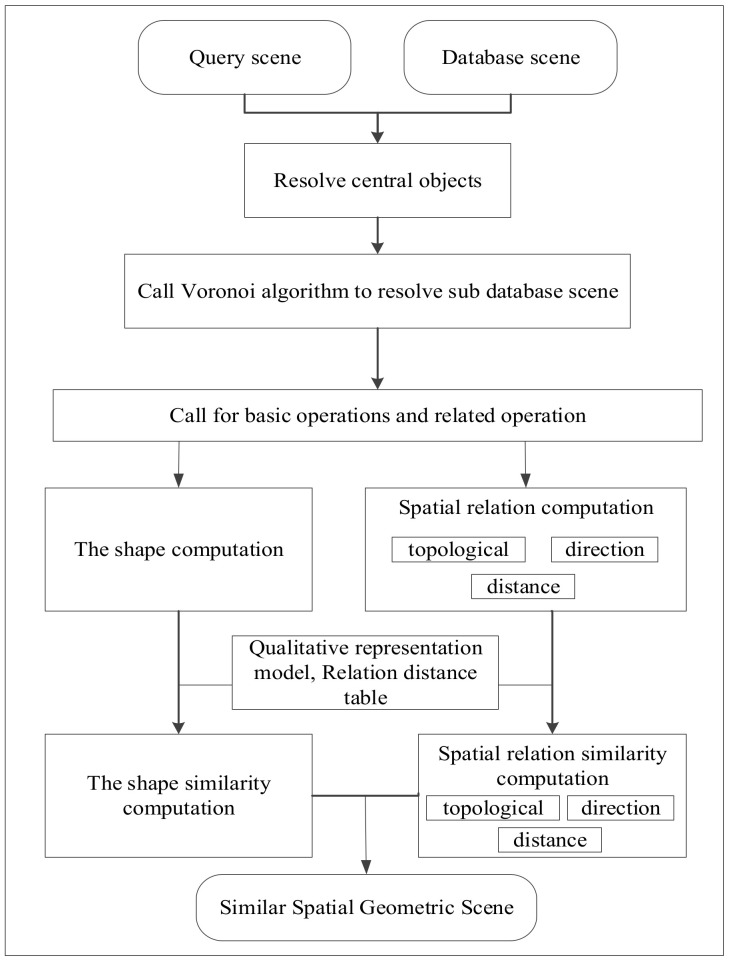
Spatial geometric similarity retrieval process.

**Figure 2 ijerph-19-10807-f002:**
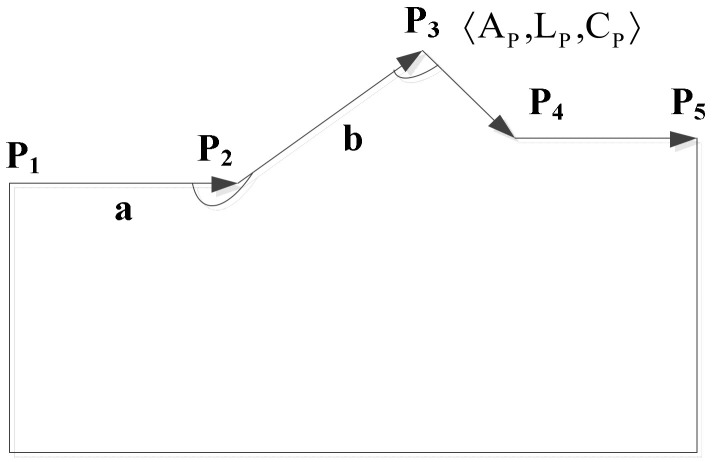
Sketch of shape parameter computation.

**Figure 3 ijerph-19-10807-f003:**
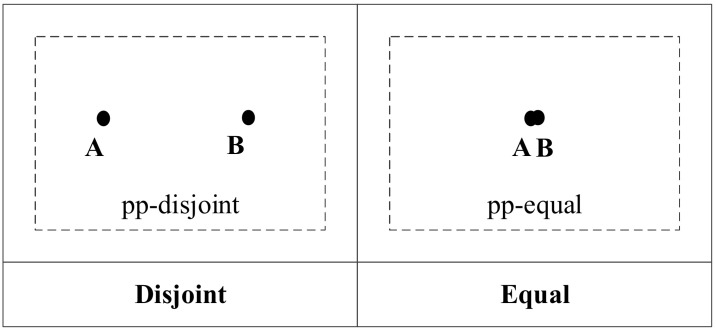
Topological relations between two points.

**Figure 4 ijerph-19-10807-f004:**
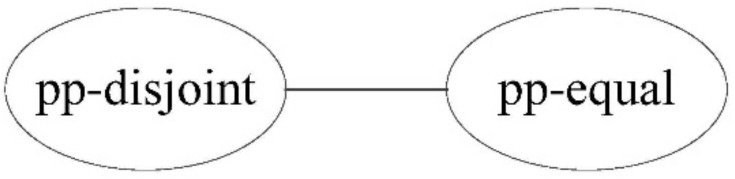
Conceptual neighborhood diagram of topological relations between two points.

**Figure 5 ijerph-19-10807-f005:**
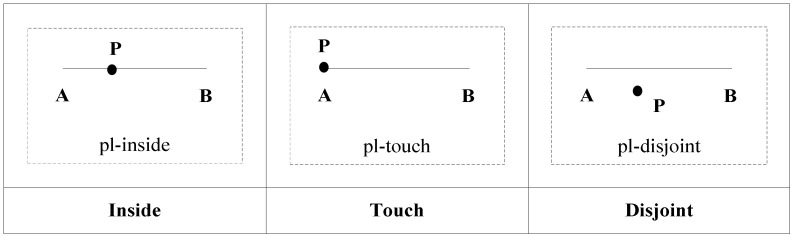
Topological relations between a point and a line segment.

**Figure 6 ijerph-19-10807-f006:**
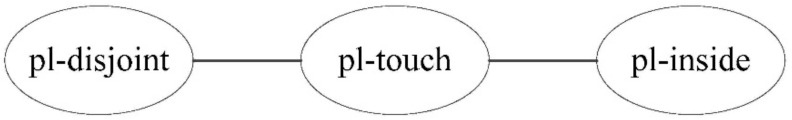
Conceptual neighborhood diagram of topological relations between a point and a line segment.

**Figure 7 ijerph-19-10807-f007:**
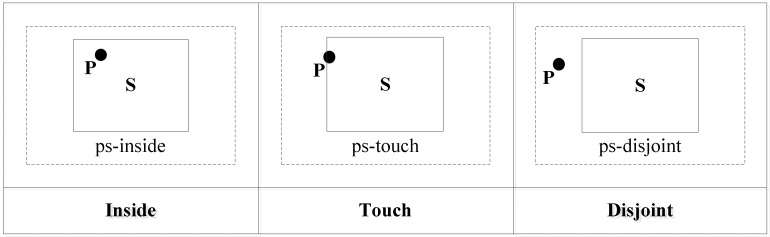
Topological relations between a point and a polygon.

**Figure 8 ijerph-19-10807-f008:**
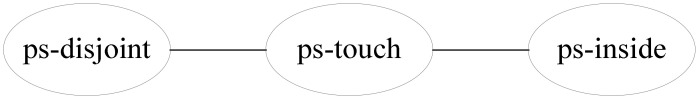
Conceptual neighborhood diagram of topological relations between a point and a polygon.

**Figure 9 ijerph-19-10807-f009:**
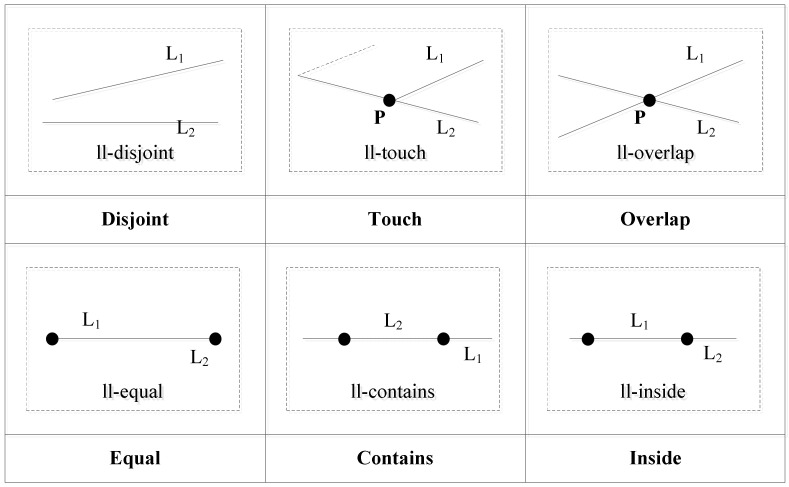
Topological relations between two line segments.

**Figure 10 ijerph-19-10807-f010:**
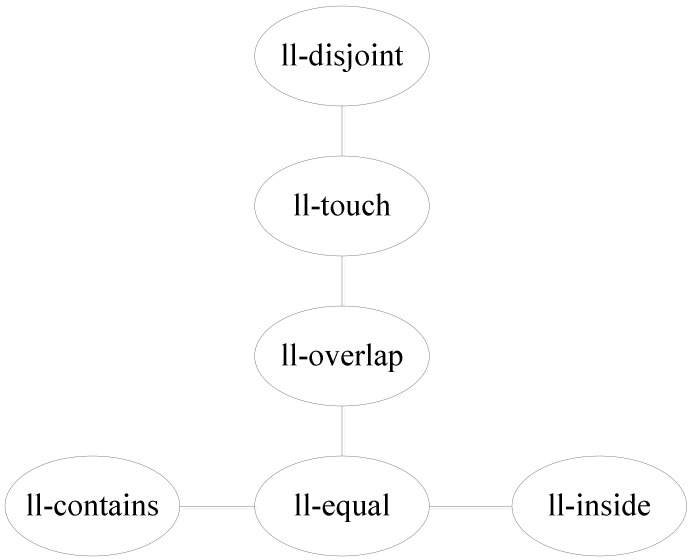
Conceptual neighborhood diagram of topological relations between two line segments.

**Figure 11 ijerph-19-10807-f011:**
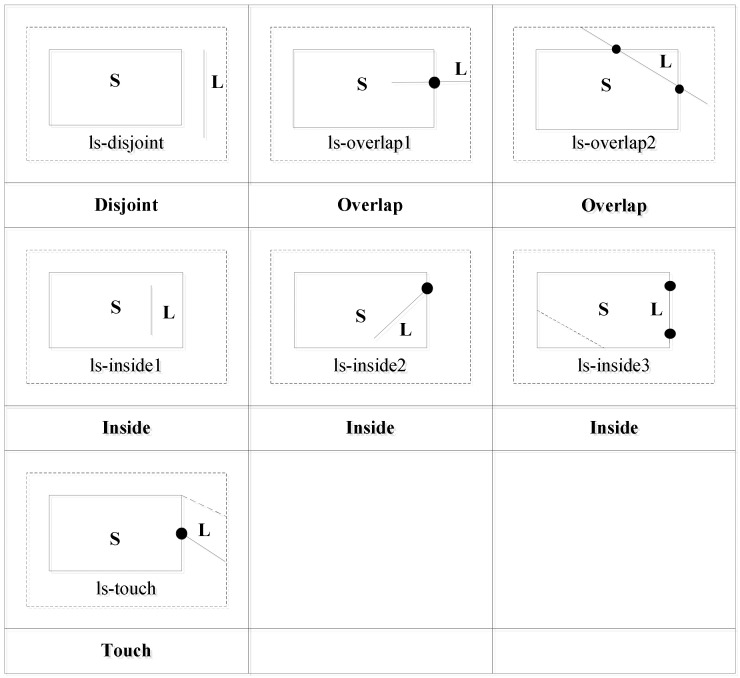
Topological relations between a line segment and a polygon.

**Figure 12 ijerph-19-10807-f012:**
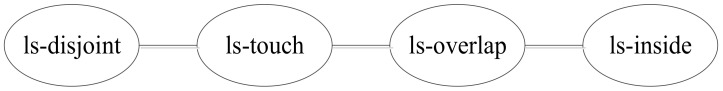
Conceptual neighborhood diagram of topological relations between a line segment and a polygon.

**Figure 13 ijerph-19-10807-f013:**
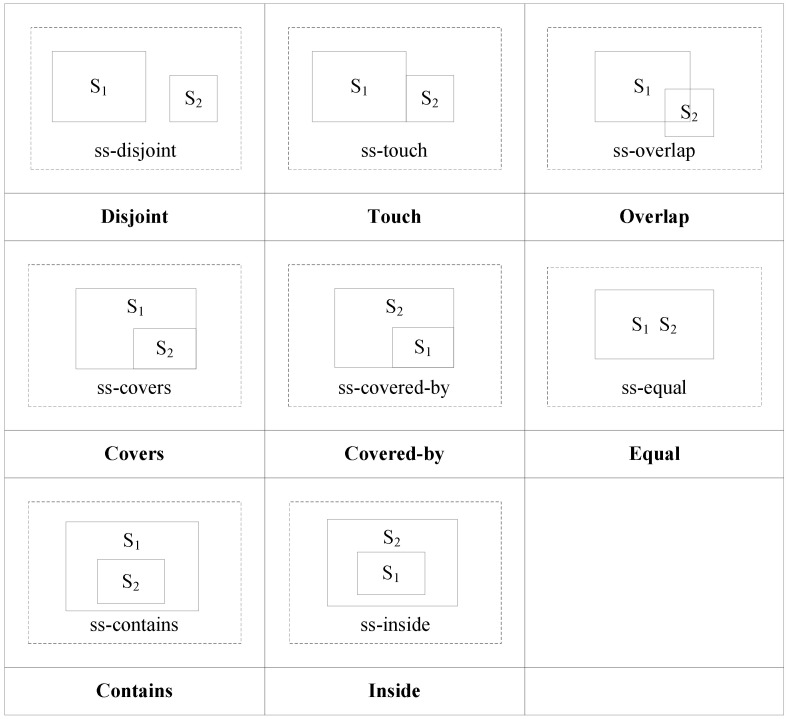
Topological relations between two polygons.

**Figure 14 ijerph-19-10807-f014:**
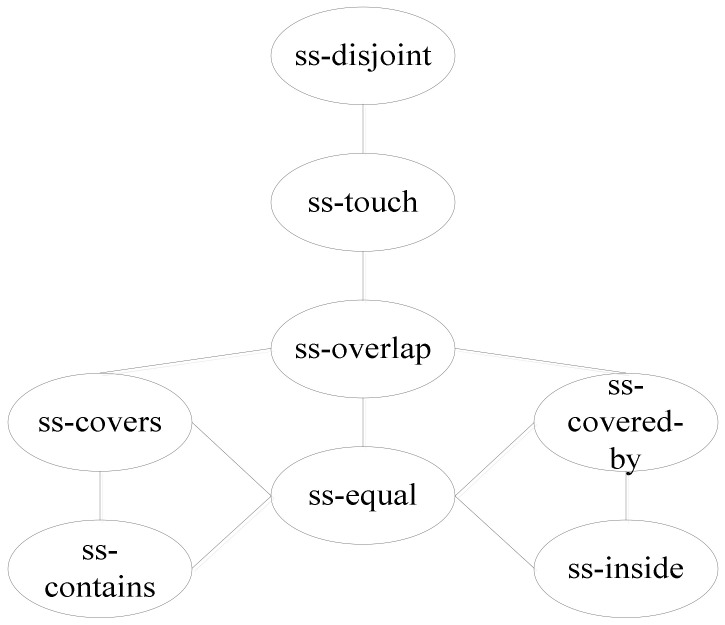
Conceptual neighborhood diagram of topological relations between two polygons.

**Figure 15 ijerph-19-10807-f015:**
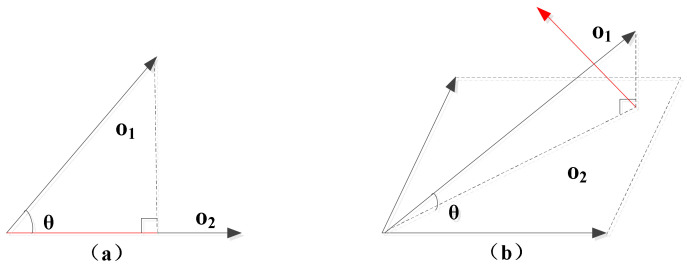
Inner product. (**a**) Inner product between two vectors; (**b**) Inner product between one vector and one plane.

**Figure 16 ijerph-19-10807-f016:**
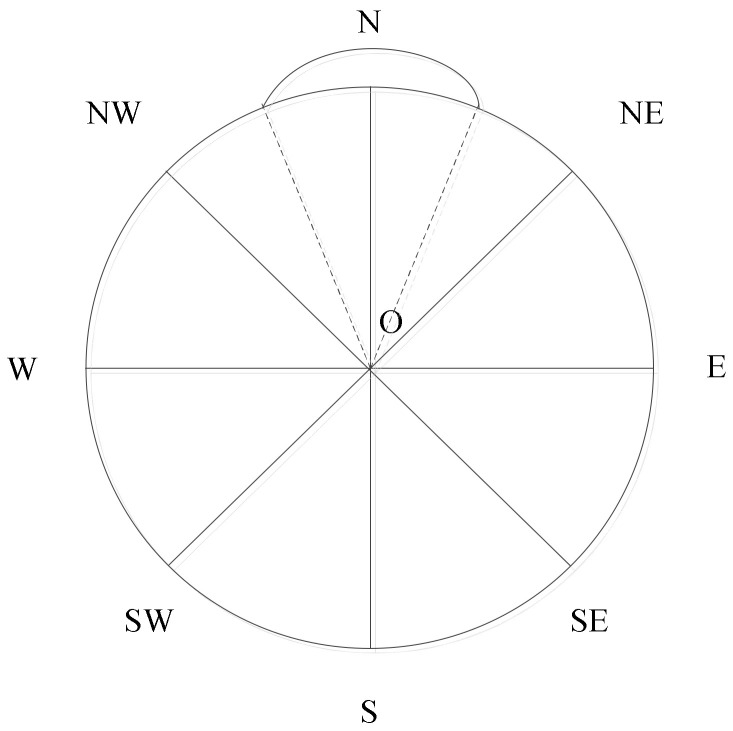
Relation graph between direction relation semantic expression and angle value.

**Figure 17 ijerph-19-10807-f017:**
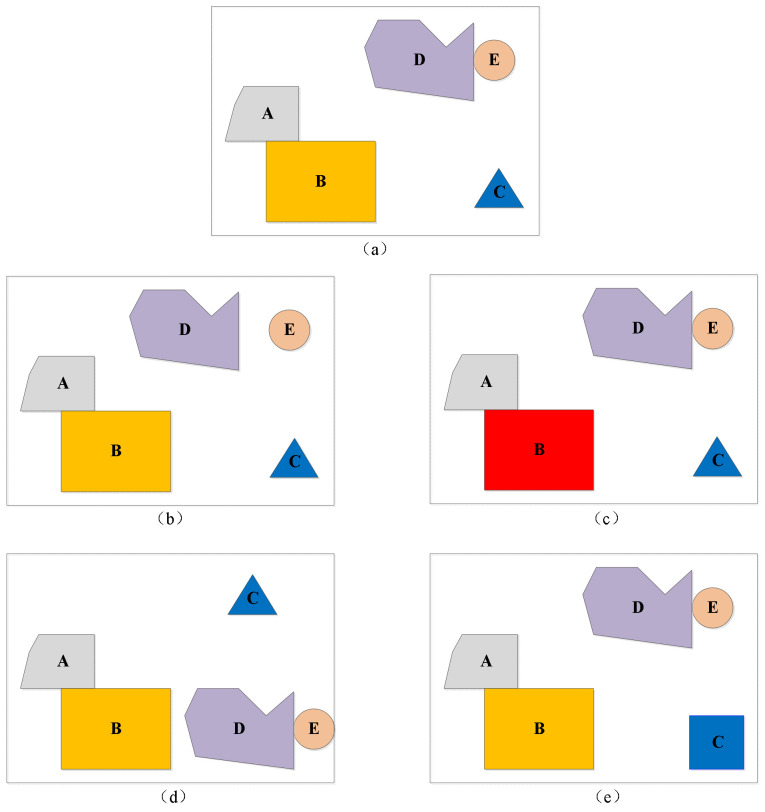
(**a**) Query scene. (**b**–**e**) Database scenes.

**Figure 18 ijerph-19-10807-f018:**
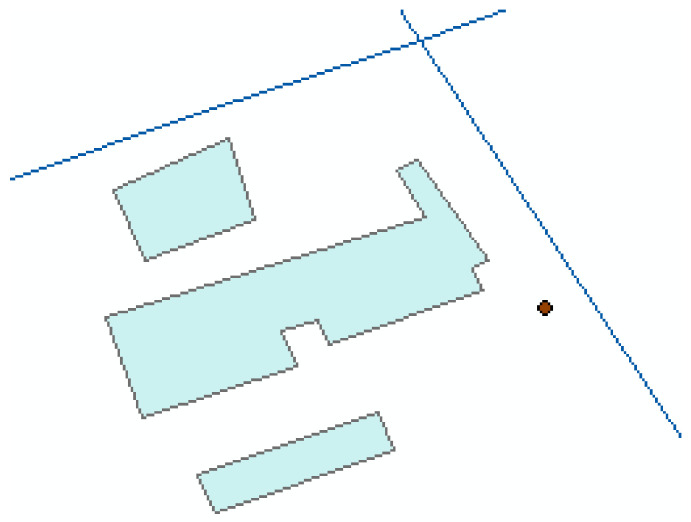
One query scene.

**Figure 19 ijerph-19-10807-f019:**
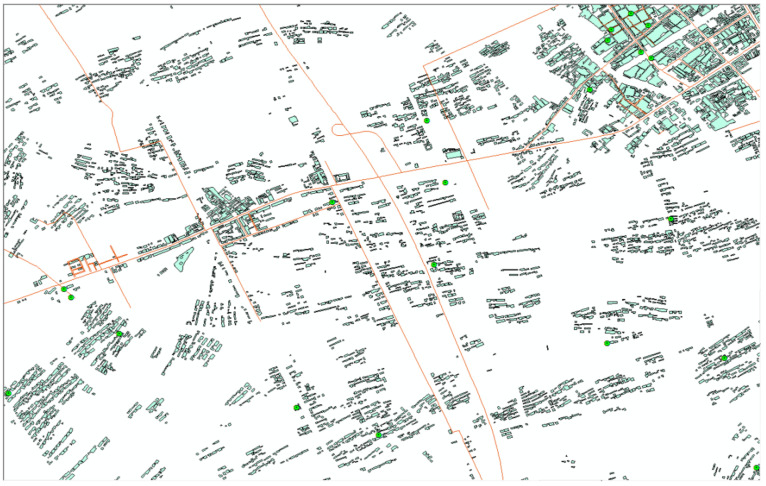
Database scene.

**Figure 20 ijerph-19-10807-f020:**
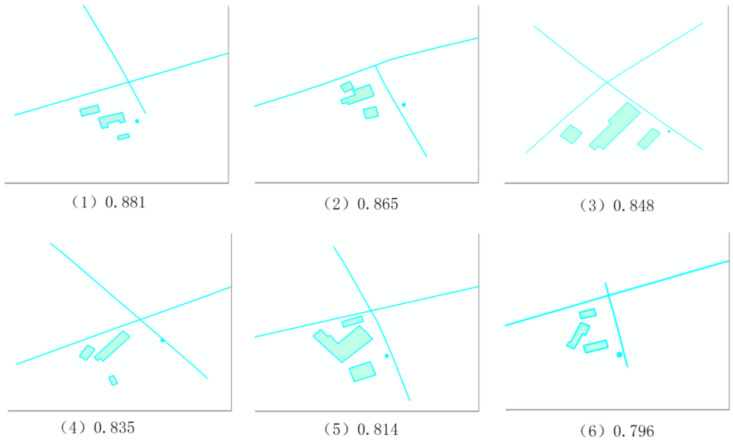
Retrieval results.

**Table 1 ijerph-19-10807-t001:** Angle relation distances.

	Very Acute	Acute	Right	Obtuse	Very Obtuse
**Very acute**	0.0	42.6	71.6	97.5	140.0
**Acute**	42.6	0.0	32.6	55.0	97.5
**Right**	71.6	32.6	0.0	32.6	71.6
**Obtuse**	97.5	55.0	32.6	0.0	42.6
**Very obtuse**	140.0	97.5	71.6	42.6	0.0

**Table 2 ijerph-19-10807-t002:** Length ratio relation distances.

	msh	hl	absh	sl	abl	dl	ml
**msh**	0.00	0.32	0.55	0.81	1.32	1.80	6.95
**hl**	0.32	0.00	0.25	0.50	1.04	1.50	6.75
**absh**	0.55	0.25	0.00	0.25	0.79	1.25	6.52
**sl**	0.81	0.50	0.25	0.00	0.58	1.00	6.35
**abl**	1.32	1.04	0.79	0.58	0.00	0.58	5.77
**dl**	1.80	1.50	1.25	1.00	0.58	0.00	5.59
**ml**	6.95	6.75	6.52	6.35	5.77	5.59	0.00

**Table 3 ijerph-19-10807-t003:** Topological relation distances between a point and a line segment.

Point–Line	pl-Disjoint	pl-Touch	pl-Inside
**pl-disjoint**	0	1	2
**pl-touch**	1	0	1
**pl-inside**	2	1	0

**Table 4 ijerph-19-10807-t004:** Topological relation distances between a point and a polygon.

Point–Polygon	ps-Disjoint	ps-Touch	ps-Inside
**ps-disjoint**	0	1	2
**ps-touch**	1	0	1
**ps-inside**	2	1	0

**Table 5 ijerph-19-10807-t005:** Topological relation distances between two line segments.

Line–Line	ll-Disjoint	ll-Touch	ll-Overlap	ll-Equal	ll-Contains	ll-Inside
**ll-disjoint**	0	1	2	3	4	4
**ll-touch**	1	0	1	2	3	3
**ll-overlap**	2	1	0	1	2	2
**ll-equal**	3	2	1	0	1	1
**ll-contains**	4	3	2	1	0	2
**ll-inside**	4	3	2	1	2	0

**Table 6 ijerph-19-10807-t006:** Topological relation distance between a line segment and a polygon.

Line–Polygon	ls-Disjoint	ls-Touch	ls-Overlap	ls-Inside
**ls-disjoint**	0	1	2	3
**ls-touch**	1	0	1	2
**ls-overlap**	2	1	0	1
**ls-inside**	3	2	1	0

**Table 7 ijerph-19-10807-t007:** Topological relation distances between a line segment and a polygon.

Polygon–Polygon	ss-Disjoint	ss-Touch	ss-Overlap	ss-Covers	ss-Covered-by	ss-Contains	ss-Inside	ss-Equal
**ss-disjoint**	0	1	2	3	3	4	4	3
**ss-touch**	1	0	1	2	2	3	3	2
**ss-overlap**	2	1	0	1	1	2	2	1
**ss-covers**	3	2	1	0	2	1	2	1
**ss-covered-by**	3	2	1	2	0	2	1	1
**ss-contains**	4	3	2	1	2	0	2	1
**ss-inside**	4	3	2	2	1	2	0	1
**ss-equal**	3	2	1	1	1	1	1	0

**Table 8 ijerph-19-10807-t008:** Direction relation distance.

	N	NE	E	SE	S	SW	W	NW
**N**	0	1	2	3	4	3	2	1
**NE**	1	0	1	2	3	4	3	2
**E**	2	1	0	1	2	3	4	3
**SE**	3	2	1	0	1	2	3	4
**S**	4	3	2	1	0	1	2	3
**SW**	3	4	3	2	1	0	1	2
**W**	2	3	4	3	2	1	0	1
**NW**	1	2	3	4	3	2	1	0

**Table 9 ijerph-19-10807-t009:** Distance relation distance.

Length × Δ	vc	cl	abc	co	abf	f	vf
**vc**	0.000	0.082	0.133	0.200	0.354	0.552	0.806
**cl**	0.082	0.000	0.051	0.122	0.279	0.475	0.730
**abc**	0.133	0.051	0.000	0.076	0.234	0.428	0.682
**co**	0.200	0.122	0.076	0.000	0.158	0.354	0.608
**abf**	0.354	0.279	0.234	0.158	0.000	0.200	0.453
**f**	0.552	0.475	0.428	0.354	0.200	0.000	0.255
**vf**	0.806	0.730	0.682	0.608	0.453	0.255	0.000

**Table 10 ijerph-19-10807-t010:** Topological relations in the query scene (a) in Figure 17.

	A_1_	B_1_	C_1_	D_1_	E_1_
**A_1_**	ss-equal	ss-touch	ss-disjoint	ss-disjoint	ss-disjoint
**B_1_**	ss-touch	ss-equal	ss-disjoint	ss-disjoint	ss-disjoint
**C_1_**	ss-disjoint	ss-disjoint	ss-equal	ss-disjoint	ss-disjoint
**D_1_**	ss-disjoint	ss-disjoint	ss-disjoint	ss-equal	ss-touch
**E_1_**	ss-disjoint	ss-disjoint	ss-disjoint	ss-touch	ss-equal

**Table 11 ijerph-19-10807-t011:** Direction relations in the query scene (a) in Figure 17.

	A_1_	B_1_	C_1_	D_1_	E_1_
**A_1_**	O	NW	W	SW	W
**B_1_**	SE	O	W	SW	SW
**C_1_**	E	E	O	SE	S
**D_1_**	NE	NE	NW	O	W
**E_1_**	E	NE	N	E	O

**Table 12 ijerph-19-10807-t012:** Distance relations in the query scene (a) in Figure 17.

	A_1_	B_1_	C_1_	D_1_	E_1_
**A_1_**	/	abf	vf	vf	vf
**B_1_**	abf	/	vf	f	vf
**C_1_**	vf	vf	/	f	f
**D_1_**	vf	f	f	/	co
**E_1_**	vf	vf	f	co	/

**Table 13 ijerph-19-10807-t013:** Topological relations in the database scene (b) in Figure 17.

	A_2_	B_2_	C_2_	D_2_	E_2_
**A_2_**	ss-equal	ss-touch	ss-disjoint	ss-disjoint	ss-disjoint
**B_2_**	ss-touch	ss-equal	ss-disjoint	ss-disjoint	ss-disjoint
**C_2_**	ss-disjoint	ss-disjoint	ss-equal	ss-disjoint	ss-disjoint
**D_2_**	ss-disjoint	ss-disjoint	ss-disjoint	ss-equal	ss-disjoint
**E_2_**	ss-disjoint	ss-disjoint	ss-disjoint	ss-disjoint	ss-equal

**Table 14 ijerph-19-10807-t014:** Direction relations in the database scene (b) in Figure 17.

	A_2_	B_2_	C_2_	D_2_	E_2_
**A_2_**	O	NW	W	SW	W
**B_2_**	SE	O	W	SW	SW
**C_2_**	E	E	O	SE	S
**D_2_**	NE	NE	NW	O	W
**E_2_**	E	NE	N	E	O

**Table 15 ijerph-19-10807-t015:** Distance relations in the database scene (b) in Figure 17.

	A_2_	B_2_	C_2_	D_2_	E_2_
**A_2_**	/	abf	vf	f	vf
**B_2_**	abf	/	vf	f	vf
**C_2_**	vf	vf	/	vf	f
**D_2_**	f	f	vf	/	abf
**E_2_**	vf	vf	f	abf	/

**Table 16 ijerph-19-10807-t016:** Topological relations in the database scene (d) in Figure 17.

	A_4_	B_4_	C_4_	D_4_	E_4_
**A_4_**	ss-equal	ss-touch	ss-disjoint	ss-disjoint	ss-disjoint
**B_4_**	ss-touch	ss-equal	ss-disjoint	ss-disjoint	ss-disjoint
**C_4_**	ss-disjoint	ss-disjoint	ss-equal	ss-disjoint	ss-disjoint
**D_4_**	ss-disjoint	ss-disjoint	ss-disjoint	ss-equal	ss-touch
**E_4_**	ss-disjoint	ss-disjoint	ss-disjoint	ss-touch	ss-equal

**Table 17 ijerph-19-10807-t017:** Direction relations in the database scene (d) in Figure 17.

	A_4_	B_4_	C_4_	D_4_	E_4_
**A_4_**	O	NW	W	W	W
**B_4_**	SE	O	NE	W	W
**C_4_**	E	SW	O	N	NW
**D_4_**	E	E	S	O	W
**E_4_**	E	E	SE	E	O

**Table 18 ijerph-19-10807-t018:** Distance relations in the database scene (d) in Figure 17.

	A_4_	B_4_	C_4_	D_4_	E_4_
**A_4_**	/	abf	vf	vf	vf
**B_4_**	abf	/	vf	abf	f
**C_4_**	vf	vf	/	abf	f
**D_4_**	vf	abf	abf	/	abf
**E_4_**	vf	f	f	abf	/

**Table 19 ijerph-19-10807-t019:** Geometric similarity values between each scene and query scene.

	S_a_-S_b_	S_a_-S_c_	S_a_-S_d_	S_a_-S_e_
**Similarity Value**	0.968	1.000	0.933	0.969

## Data Availability

The data is not available due to legal restrictions. The authors of this study did not agree for their data to be shared publicly, so supporting data is not available.

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
