# Peer review of "Study on Spatial Geometric Similarity Based on Conformal Geometric Algebra"

_ijerph, 2022, doi:10.3390/ijerph191710807_

Round 1

Reviewer 1 Report

The paper explains the method to compute spatial scene similarity using CGA. First the object shapes and their spatial relations are constructed using the inner, outer and geometric products in CGA. Then relation distance tables are created by using conceptual neighborhood diagrams and interval distances. Based on the objects shape and spatial relations, the similarity between two objects is computed. To make it comprehensive, all the aspects like spatial object shape, spatial topology, direction and distance between two objects is also taken into account.

Points of critique:

1. The relation tables need more explanation like the use of identical names for the x and y axis and how the relation is decided. It is not clear whether this relationship is assigned based on properties or computed by some numerical method.

2. What is the "Borelian notation"? Something related to Borel algebra? If so, introduce it and provide a citation. 

3. All tables need captions both in methodology and also in results.

4. Sec 3.2 only mentions that this study is based on the measurement model from the cited paper

   and then proceeds to list the components of measurement model. I think there should be some context

   explaning the model and the choice of components.

5. Line 45: "In 2001, Chinese mathematician Li Hongbo proposed the theory of Conformal

   Geometric Algebra (CGA), and it was introduced to geoscience field". Citation?

   The work by L.Dorst (Amsterdam), J. & A. Lasenby (Cambridge), D. Hildenbrand (Dortmund) and particularly D. Hestenes in the 1960s are entirely ignored in this paper. Find and include appropriate references.

6. Line 133 they say they use the inner product to compute the angle. That is nothing that specifically requires GA and is actually a bad idea to do it that way because it only yields the cosine of the angle from the inner product, with limited sign and range. This requires then the presented hack.

   Whereas the geometric product encodes the inner and the outer product - the inner product is proportional to the cosine, the outer product is the sine. Both together gives the arctan2 function which is unique for the entire range - so actually using the full geometric product gives the full range of the correct angle, and in an article about geometric algebra it is surprising that this method is not used.

7. As per section 3.4, it needs to be clarified that the presented algorithm does not only refer to vectors, but to geometrical primitives as described by CGA, otherwise this section is trivial. A recommendable reference here is "Analyzing the inner product of 2 circles with Gaalop" -  https://doi.org/10.1002/mma.4471 . Put this your section in context with the analysis presented there for better illustration of this core algorithm.

8. Table 3 doesn't make sense as it is. Same names on each axis makes the table meaningless (besides, avoid the page break between table header and table content). The table as is does not convey any information.

   There are similar issues with the other tables.

9.  It is not clear how to extract the essence of the whole paper in terms of methodology. It should be more clear to grasp.

10. It may help if fig 16 is placed be at the beginning of Methodology and the sections should be briefed here  and then the explanation should start.

Reviewer 2 Report

Spatial scene similarity based on conformal geometric algebra was presented in this article. Spatial relation similarity and shape similarity are considered in this article. Some specific suggestions and comments are as follows.

(1) Literature Review: Spatial scene similarity are not only related to the geometric characteristics of spatial objects and spatial relations, but also closely related to semantics or attributes. Related literature on semantic or attribute similarity should be added.

(2) Literature Review: What is a space scene? Please give an explanation or definition. The current progress of space scene research should be added.

(3) Formula 6: Please explain further how to calculate WRi?

(4) Section 3.4: How to calculate the direction relation between non-point objects?

(5) Section 3.5: How to calculate the distance relation between non-point objects?

(6) Section 3.6: How to calculate the shape similarity?  

(7) Line 406: Shape similarity alone is not enough. Moreover, the number of spatial objects in a spatial scene may be large, and if the shape similarity of all objects is calculated once, this algorithm may take a long time to execute.

(8) The semantics or attributes in spatial scene similarity are quite important and cannot be ignored. 

Round 2

Reviewer 2 Report

The semantics or attributes in spatial scene similarity are quite important and cannot be ignored. The authors did not consider the semantics or attributes of spatial scene similarity in the revised paper.
